# Social Insurance Physician Burnout—Stress Factors and Coping Strategies

**DOI:** 10.3390/medicina59030436

**Published:** 2023-02-22

**Authors:** Corina Oancea, Anicuta Cernamoriti, Despina Mihaela Gherman, Florina Georgeta Popescu

**Affiliations:** 1Department of Physical Medicine and Rehabilitation, Carol Davila University of Medicine and Pharmacy, 020021 Bucharest, Romania; 2The National Institute for Medical Assessment and Work Capacity Rehabilitation, 050659 Bucharest, Romania; 3Department of Internal Medicine, Victor Babes University of Medicine and Pharmacy, 300041 Timisoara, Romania

**Keywords:** social insurance, physician, professional burnout, risk factors, coping behaviors

## Abstract

*Background and Objective* Burnout syndrome is well-documented and highly prevalent among healthcare professionals. The literature search found studies mainly aimed at front-line medical specialties, cardiologists, or physicians working in intensive care units. Workload and work conditions favor the occurrence of burnout syndrome among social insurance physicians, with many consequences on health status and a decrease in the quality of their work. We aimed to assess the degree of vulnerability to developing burnout syndrome, factors associated with stress, and coping strategies at social insurance physicians. Materials and *Methods:* Social insurance physicians working in territorial services for medical assessment of work capacity from Romania participated in the study. An observational study was performed to describe the extent of the exhaustion syndrome among social insurance physicians (SIPhs). Three questionnaires were filled out by the participants: a short version of MBI-HSS to analyze the degree of burnout, an interview with specific questions for the source of stress and Brief-COPE for stress control. Brief demographic data were also collected. Data were statistically analyzed with appropriate tests using PSPP software. *Results:* Seventy-four physicians were included in the study. Fifty-six were females (75.7%) and twenty-eight (38%) had moderate or high burnout and cognitive distortions with depression resulting as a major side-effect (*p* < 0.001). Professional factors, mainly deadline pressure (*p* < 0.001) and high workload (*p* = 0.012), have emerged as contributing factors to burnout syndrome. Mental disengagement (*p* = 0.001), active coping (*p* = 0.006), and acceptance (*p* = 0.014) would improve stress control. *Conclusion:* More than two-thirds of social insurance physicians had moderate and high burnout syndrome. The development of strategies to standardize workload was identified as an important action area, along with the long-term preservation of health status and professional performance.

## 1. Background

Burnout is defined as a psychological syndrome resulting from chronic workplace stress that has not been successfully managed, characterized by exhaustion, cynicism, and reduced professional efficiency. World Health Organization gives increasing importance to the occurrence of burnout syndrome among employees. In 2019, the 72nd World Health Assembly introduced it as an occupational phenomenon for the 11th revision that takes effect in 2022 [1].

Burnout syndrome is well-documented and highly prevalent among healthcare professionals [2]. Studies reported figures of up to 80% of physicians, with substantial variability in prevalence, as showed by Rotenstein et al. in a systematic review [3]. Differences in study designs, burnout definition, and assessment methods were also found. This information highlights the impossibility of obtaining clear figures on the prevalence of burnout and emphasizes the importance of reaching an agreement on the definition of burnout and the standardization of measurement tools to assess the effects of chronic occupational stress on physicians [3].

It is mainly described in front-line medical specialties, cardiologists [4], emergency physicians [5,6], or physicians working in intensive care units [7,8]. More than one-third of primary care practitioners stated that they felt exhausted [9,10,11,12].

Social insurance physicians or medical advisers are physicians specialized in social security, which assess the eligibility for work disability pension and other social insurance rights [13,14]. Their activity involves mainly “in person” evaluations and requires good quality evaluation, attention to claimants’ satisfaction, but also fair examination reported to the administrative institutions and state budget.

Dealing with pressure from both sides, insurers and clients, and the need for an accurate evaluation, given that decisions turn into public money expenses, may trigger burnout among social insurance physicians. This results in long-term health problems and decreased performance.

Physician burnout is linked with low job satisfaction and may have negative consequences also to the healthcare system as a whole. Career dissatisfaction in physicians is linked with leaving medicine, thus, contributing to workforce shortages [4]. The loss of physicians through career dropout, reduced work, or early retirement [15] is a phenomenon also occurring among SIPhs. This disrupts workflows and reduces efficiency, which is detrimental to the good functioning of medical assessment services.

In European social security organizations, several countries have reported shortages of social insurance physicians [16], and Romania is no exception. According to the national records, around 30% of SIPhs have left the national network or reduced their activity in the last 15 years.

With a significant number of physicians leaving the system, the amount of work has been transferred and increased for those who remain until a viable long-term solution is found, including attracting other specialists into the system. Several hypotheses have been discussed for this lack. Like their European colleagues, Romanian SIPhs frequently reported organizational problems and computer system deficiencies. Other aspects were also considered besides the increased demand for medical evaluations: a greater focus on professional reintegration; case assessments became increasingly demanding and time-consuming due to the shift to more complex or migrant cases [17].

The medical assessments for short-term (sickness) and long-term (disability) benefits have traditionally been undertaken by social insurance physicians [18]. Other tasks were also described and these vary considerably between countries. In Romania, SIPhs also coordinate the return to work program, and some participate in appeals commissions [14].

From a different perspective, effective handling of stress improves quality of work life (QWL) important factor which is tied to satisfaction at the workplace and better results. According to different authors, the quality of work life is divided into the work environment, work conditions, organizational and interpersonal relationships, occupational stress, possibilities of professional development, wages, and social support [19].

Job satisfaction was found as a negative predictor of each type of burnout subscale by some authors, while the conflict was a factor contributing positively both to emotional exhaustion and depersonalization scores [20].

Good QWL increases the level of employee retention. The result is for the benefit of both the employees and the institution’s long-term viability [21].

The most widely used measure of burnout is the Maslach Burnout Inventory, which has high reliability and validity [22]. As shown by Rotenstein et al. in a systematic review, 85.7% of studies used a version of the Maslach Burnout Inventory to assess burnout [3]. However, there are different versions of this scale. MBI-HSS was designed to assess the intensity of perceived burnout among professionals active in human services. The MBI-HSS assesses all three dimensions of burnout: emotional exhaustion (feeling emotionally overextended by work), depersonalization (an unfeeling and impersonal response to people), and personal accomplishment (feelings of incompetence and lack of achievement at work) [2]. The MBI-HSS is translated, validated [23], and widely used [24] in Romania in different studies, and some researchers made individual analyses of its items [25] or used adapted MBI-HSS [26,27].

Coping strategies play an important role in the process of adapting to challenging conditions. Coping is defined as dealing with stressful factors and events and the reactions of individuals in an attempt to overcome these situations [28]. Appropriate coping strategies may increase individual resistance and protect from developing mental disorders [29]. Conversely, an inappropriate way to deal with high-level stress is for employees to psychologically withdraw from others, leading to dehumanized and depersonalized contact. This dysfunctional coping strategy will lead to further deterioration of the relationship with others [30,31]. Each person may act differently depending on various factors such as age, gender, cultural background, or personality traits [32].

COPE questionnaire (Coping Orientation for Problem Experiences Scale) was designed to measure the different coping strategies people use in response to stress. COPE is a self-reporting scale consisting of 60 questions and 15 subscales. Each subscale contains four questions and provides information on separate coping skills. In conclusion, the scores obtained in different subscales allow interpretation of which coping strategy is predominantly used by the individuals. The subscales measure problem-based methods (including planning, use of instrumental social support, active coping, restraint, suppression of competing activities), emotion-based methods (including use of emotional, social support, positive reinterpretation and growth, religious coping, humor, and acceptance) or non-functional coping strategies (venting of emotions, denial, behavioral disengagement, mental disengagement, and substance use) [33].

## 2. Objective

The main objective of this study was to assess the degree of vulnerability to developing burnout syndrome, factors associated with stress, and coping strategies of social insurance physicians. To our knowledge, no other study has investigated burnout in this particular category of healthcare professionals.

In this paper, the following research questions were addressed: What is the prevalence of different components of burnout? Which are predictive factors for burnout? Which are burnout side-effects? Which are the stress factors? Which are the coping strategies used for stress control?

## 3. Methods

An observational study was performed to describe the extent of the exhaustion syndrome among social insurance physicians. This type of study was considered the most appropriate considering the low cost and possibility of analyzing a well-defined population in a relatively short period of time. Questionnaires were sent by email to eliminate the respondents’ reluctance to be evaluated by the research team. We took into consideration also other advantages: fast sending of questionnaires, the integrity of data obtained, flexibility, and low costs because it does not require printing, multiplication, or travel expenses for the operator.

We aimed to obtain the most accurate answers, so through an information letter, it was explained to the participants that the study was conducted for research purposes. The medical staff were assured of their privacy being respected, and responses were analyzed anonymously.

Some sociodemographic data were collected: age, gender, marital status, children, and length of service duration. The participants also reported on: vacations, sick leave, psycho-traumatic events, daily commutes, or workload. We considered psycho-trauma a stressful, disruptive event such as the death of a close relative, divorce, change of city residence, job change, or major workplace conflict.

The shortened version of the scale MBI-HSS translated into Romanian was used to analyze burnout syndrome components [34]. Two items were excluded from the original scale, 12 (“I feel very energetic”) and 16 (“Working directly with people puts too much stress on me”), as recommended by several authors to obtain a more valid tool [35,36,37,38,39].

This 20-item scale consists of 8 items for nervous exhaustion, 5 items for depersonalization, and 7 items for personal accomplishment. Burnout is identified by high scores at nervous exhaustion and depersonalization and low scores at personal accomplishment. The answers are based on the frequency the subject experiences the respective feelings, from a range from 0 (never) to 6 (every day). The scores obtained for each dimension are then classified into three categories: low, moderate, and high.

Coping mechanisms were analyzed using a 28-item Brief-COPE with questions translated into Romanian [40]. Brief COPE is considered an efficient tool to evaluate various coping strategies, having adequate psychometric properties. Different ways of coping with stress are addressed, and all the items are rated on a 4-point scale ranging from 1—usually, I do not do this at all to 4—usually, I do this a lot. This instrument was subsequently analyzed in detail. First, 14 subscales were used, composed of two items each: active coping, planning, suppression of competing activities, restraint from action, seeking instrumental social support, seeking emotional, social support, positive reinterpretation, acceptance, denial, emotional discharge, religious coping, mental disengagement, behavioral disengagement, and substance abuse.

Further, following Carver’s approach, we grouped three different categories as problem-based methods (active coping, planning, suppression of competing activities, restraint from action, seeking instrumental social support), emotion-based methods (seeking emotional, social support, positive reinterpretation, religious coping, acceptance) and non-functional coping strategies (denial, emotional discharge, mental disengagement, behavioral disengagement, and substance abuse) [33].

The study was carried out with the aim of characterizing this population group and testing the possible response rate by reducing the number of questions and offering anonymity.

Following a comprehensive evaluation and validation by statistical analysis, we proposed a questionnaire-based survey to evaluate stressors and coping strategies for social insurance physicians, with the following items: sociodemographic data, short MBI-HSS, brief COPE, and a set of questions focused on the potential inducing factors for burnout. Thus, a complex assessment tool was obtained, easy to apply, and with reduced costs.

## 4. Statistical Analysis

All parameters were compared between the group with and without burnout syndrome by chi-square test for nominal and categorical variables (gender, marital status, children, psycho-traumatic events, daily commute or workload) or by independent-sample *t*-test for numerical variables (e.g., age, length of service duration, number of days of vacation, number of days of sick leave). Multiple logistic regression was used to obtain odds ratios and 95% confidence intervals for each variable to be assessed as a potential predictor for burnout syndrome. The type of association of different parameters with the presence of burnout components was studied by correlation analysis. All statistical analyses were performed by PSPP software.

## 5. Ethical Statement

The study was approved by the Scientific Research Ethics Committee of the Carol Davila University of Medicine and Pharmacy Bucharest. This study complied with the World Medical Association’s Declaration of Helsinki. Participation in this survey was regarded as agreeing to the study, and all data were de-identified.

## 6. Results

### Participants

Seventy-four social insurance physicians working in territorial services for medical assessment of work capacity from all over the country voluntarily participated in the study. The 49% survey response rate was considered satisfactory for the results to be representative of this professional category.

Stage 1. Burnout syndrome using the short version of MBI-HSS
Nervous exhaustion
Prevalence

Females were 56 (75.7%) of the responders. The average age of participants was 48.86 years old. Of the responders, 81.08% were in a couple, and 78.38% had children. All participants were working in territorial medical services of the National House of Pensions with an average length of service of 17.09 years, and 35.14% reported having a second job. Twenty-four participants used sick leave in the last year with a mean duration of 5.95 days, and 72 participants took vacations for an average of 18.43 days.

Twenty-six respondents experienced at least one psycho-traumatic event in the past year. About one-third of participants do not work in their place of residence, and three-quarters reported having high or excessive workloads. The sociodemographic and professional characteristics of participants are summarized in Table 1.

Of the 74 respondents, 28 respondents (38%) suffered from moderate or high burnout. The average scores for this population were 18.64(± 4.75) vs. 6.43 (± 4.67); *p* < 0.001. 

      ii.Socio-demographic risk factors for burnout

We investigated the relationship between burnout and different factors (age, gender, marital status, children, length of service, workload, daily commute, second job, psychodramatic events, sick leave, and days off).

Persons with burnout had a significantly lower number of days off (15.07 ± 7.91 vs. 20.48 ± 8.22 days; *p* = 0.007). Of 14 singles, 10 (71.43%) had burnout (*p* = 0.005). Of 48 persons without psycho-traumatic events, 34 (70.83%) did not develop burnout syndrome (*p* = 0.033). Correlation analysis showed a negative correlation between the number of days off and the degree of burnout syndrome in the sense that if the number of days off decreases, the degree of nervous exhaustion increases (r = −0.31, *p* = 0.007), the same for being in a relationship (r = −0.33, *p* = 0.004) and a positive correlation between psycho-traumatic experiences and the degree of burnout syndrome in the sense that psycho-traumatic experiences worsen the degree of nervous exhaustion (r = 0.24, *p* = 0.037). Therefore, being single, having experienced psychotraumas, and having fewer vacations appear as contributing factors to developing burnout syndrome (Table 2).

These variables were also demonstrated by multivariate logistic regression (Table 3). Several variables were significantly associated with burnout syndrome: being single (*p* = 0.036), psycho-traumatic experiences (*p* = 0.041), and a lower number of days off (*p* = 0.023).

The backward selection type of multiple logistic regression revealed that only 3 out of 11 socio-demographic variables taken into account could be considered predictive factors for burnout. In conclusion, the participants more vulnerable to burnout seemed to be single with psychological trauma and who took less time off.

      iii.Burnout side-effects

Participants with burnout experienced hostility (*p* = 0.010), professional routine (*p* < 0.001), lack of career development (*p* < 0.001), inefficiency (*p* < 0.001), reduced organization (*p* < 0.001), altered family relationships (*p* = 0.016), cognitive distortions with depression (*p* = 0.002) and poor performance at work (*p* < 0.001). The burnout indices are presented in Table 4.

    B.Depersonalization or professional detachment was analyzed as part of the questionnaire. It refers to the adoption of negative and indifferent attitudes toward others. A higher rate of average and high depersonalization (35%) was found (Table 5).

A positive correlation was obtained between depersonalization and gender and having children, and a negative one was found for daily commute. This means that being a man and having children predispose to depersonalization; instead, the daily commute was found to favor the absence of depersonalization (Table 6).

The backward selection type of multiple logistic regression retained only two of these variables to be taken into account as predictive factors for depersonalization. As a result, having children seems to make persons more vulnerable to depersonalization, probably by focusing on their own family wellness, while daily commute appears as a protective factor (Table 7).

    C.Personal accomplishment

It was also analyzed as part of the short version of MBI-HSS. All participants had low personal accomplishment with a mean of 11.62 (±6.13). 

Stage 2. Personal and professional risk factors for burnout syndrome that emerged from the interview

During the interview, the stressful situations that may contribute to the onset of burnout were investigated. Professional factors (taking on additional tasks, deadline pressure, high workload, low efficiency), as well as personal factors (extended family, personal surroundings), have been identified as potential inducing factors for burnout. (Table 8).

The backward selection type of multiple logistic regression highlighted that only two of the six factors taken into account were ascertained as the strongest contributing factors for burnout. In conclusion, professional factors, mainly deadline pressure and high workload, might have a greater potential to induce burnout than personal ones (Table 9).

Stage 3. Stress control using Brief-COPE

We investigated possible strategies adopted by the participants to manage stress effectively. Planning, active coping, emotional and social support, acceptance, mental disengagement, and emotional discharge were found to improve stress control. Stress control strategies are presented in Table 10.

Using the backward selection type of multiple logistic regression, mental disengagement through other activities (*p* = 0.023), active coping by taking action (*p* = 0.041), and accepting the reality (*p* = 0.001) has emerged as the most effective attitudes for dealing with burnout (Table 11).

The gender analysis of coping mechanisms found that men preferred problem-based methods, while for women, even if this type was also dominant, they made use of a high percentage of the other forms as well (Table 12).

## 7. Discussion

Physicians constantly suffer from distress due to burnout, which may affect their daily activity and performance [41]. This study examined markers of social insurance physicians’ well-being. We found that in 38% of cases, there were average and high levels of nervous exhaustion; in 35%, there were average and high levels of depersonalization, and all respondents had low personal accomplishment.

When comparing with similar studies using MBI-HSS, it is important to highlight the variability that exists around prevalence rates, how burnout measurement cut-off points are utilized, and how total scores and subscale results are reported [22]. In a systematic review made by Rotenstein et al., emotional exhaustion, depersonalization, and low personal accomplishment prevalence ranged from 0% to 86.2%, 0% to 89.9%, and 0% to 87.1%, respectively [3].

When looking separately at the three dimensions of burnout, low personal accomplishment was found surprisingly in all participants. Other researchers also have found high rates (from 49% to 66.9%) of low personal accomplishment among non-therapeutic medical specialties (e.g., radiologists or occupational physicians), and it was associated with low job satisfaction [42,43].

Social insurance physicians like radiologists or occupational physicians do not benefit from positive feedback from patients; on the contrary, the feedback from clients is often negative when the evaluation shows that the applicant cannot benefit from a social insurance right. While doing their job, social insurance physicians have to deal with clients’ requests, sometimes insufficient data, or the permanent requirement to make a decision by a set deadline. 

Like occupational physicians, social insurance physicians are frequently undervalued among other specialists and do not have well-established communication procedures with colleagues from other specialties or administrative staff. It was appreciated that these characteristics might be detrimental to their self-esteem. As pointed out by other researchers, the threat to identity and to self-esteem is the key to burnout syndrome in some medical specialties [43]. 

In our study, those unmarried who experienced psychotraumas or took fewer vacations were prone to develop burnout syndrome. Specific risk factors were associated with burnout in the literature: gender (female), marital status (single), age (older), or job demands (excessive) [44,45,46]. Older age [22,42] and being single [42] were reported as independent risk factors for high emotional exhaustion scores also by other researchers.

Depressive symptoms have positively correlated with burnout (r = 0.38; *p* = 0.001). Different studies showed an increased risk of depression for physicians compared to the general population. Occupational stress was associated with depression; besides personality factors, work-related factors were relevant [47].

Panagioti et al. reported difficulties in balancing work and personal life as an important factor contributing to burnout in physicians. Prioritizing work over personal time has been associated with burnout in physicians, especially women. This can lead to difficulties in maintaining personal relationships and enjoying a fulfilling private life [4].

When subjects were asked about adverse situations, professional rather than personal factors were incriminated as possible inducers of burnout. There is pressure for social insurance physicians to see more patients in shorter visits and to work a long time. In addition to seeing patients, they have to take on administrative duties and deal with increased computerization of their activities (registration of medical files and issues of various documents on the computer). These tasks significantly burden social security physicians in terms of energy demands and time management. Like other professionals, social insurance physicians need to continually update their medical knowledge and changes in social insurance legislation. The excessive workload was described as making the greatest contribution to overall job stress in other papers [4,42,48,49].

Physician’s burnout is particularly dangerous and should be avoided since it can increase medical errors and reduce the quality of care for patients [29]. Maintaining health and safety in the workplace is an important occupational health issue, and, in this regard, early evaluation of risk factors for burnout allows one to take preventive measures [50].

Interventions for burnout can be classified into two main categories, physician-directed interventions targeting individuals and organization-directed interventions targeting the working environment. Overall, physician-level interventions have been more frequently evaluated compared to organization-directed interventions. The most likely reason is that burnout was primarily viewed as a personal problem of physicians; only recently, a shift has been endeavored to view burnout as a problem of the health care organization driven by workplace factors. Two meta-analyses have suggested that both physician-directed and organization-directed interventions are effective for reducing burnout in physicians, but organization-level interventions are associated with greater improvements [51].

Physician well-being must be seen as the shared responsibility of healthcare systems and individuals. The system contribution should involve firm reforms such as the reduction of administrative tasks for physicians and increases in non-physician support staff. On the other hand, social insurance physicians should be aware of the importance of life-work balance and help-seeking for mental health problems.

Modifying changeable risk factors (mainly professional factors such as deadline pressure and high workload) and promoting professional help would ensure strong mental health for professionals. These aspects should also be taken into account by the managers of the institutions, knowing that good quality of work life increases engagement, efficiency, and results of social insurance physicians, thus, contributing as important players to a well-functioning social security system.

The gender analysis of coping mechanisms found that men preferred planning and active coping, while for women, even if the same mechanisms were dominant, their preferences were spread across several types. Our findings are consistent with other existing studies on coping and stress among physicians. Popa-Velea et al. found that women made use more than men of social support (be it emotional or instrumental), positive reframing, planning, and religious coping. Virtually, all of these variables had a buffering effect against stress, suggesting that females may benefit from a wider array of tools to fight or prevent burnout compared to men [49]. However, data about gender differences are divided. Some studies reported differences between men and women in coping strategies, while no significant differences between age or gender subgroups and coping mechanisms were reported by others [52].

Other studies investigated coping strategies using a brief-COPE questionnaire among physicians [52,53,54]. Attitudes adopted to cope with stress are very diverse, and many factors should be considered. Gender, age, personality traits, religion, geographic region of origin, or cultural specificity were cited [55].

There were described relationships between culturally learned values, beliefs, attitudes, and consecutive behavioral patterns. This concept refers to the fact that individuals operate and behave accordingly to traditionally dictated attitudes and beliefs about how life should be lived, including the manner in which one copes with life’s problems [44]. Cultural norms and family characteristics influence their ability to function. Traditional large families of certain ethnic communities facilitate seeking social support from extended family members [44], while in societies strongly influenced by religion, this becomes the most frequently used stress-coping strategy [43]. Active coping, planning, and acceptance were reported as the principal coping strategies in other studies [52,54].

From all ways of dealing with burnout, one draws particular attention in our study: the use of instrumental support from the larger category of problem-focused methods as this includes seeking medical care. The majority of persons with moderate nervous exhaustion did not try to get advice from a mental health professional. Only persons with severe form have made this step; three-quarters of women and all men did not seek professional advice. These findings might be linked to the widespread traditional social attitude of avoiding asking for specialized advice and handling the situation alone. Moreover, in Romanian society, there still is a strong current of the traditional, authoritarian-patriarchal family/traditional educational values, where the father/man is the master of the house, making the final decisions, naturally endowed with physical and intellectual strength and resistance.

Most probably, avoiding seeking treatment is due to concerns about social stigma and discrimination, but the lack of information from reliable sources often leads to less effective coping mechanisms.

Other studies reported low percentages of physicians getting professional help as a helpful coping way to make them feel less stressed. Key reasons for avoiding professional care, especially psychological ones among physicians, were cited: the belief they could manage independently, time, confidentiality, license issues, and embarrassment [46,54].

These findings are expected to draw attention to the importance of psychological support for SIPhs and their early referral to specialized services when the first signs of nervous overload are detected.

The current research represents an important step in a strategic approach to dealing with burnout in the case of social insurance physicians. The ability to identify developing problems early on, before they become more serious, can enable timely, preventive solutions. It points to the possibility of being able to customize interventions to this specific category. The potential power of this approach rests on the fact that it can function like an organizational “checkup” with repeated assessments on a regular basis.

Future management and health policy improvements are expected, and managers of the institutions and occupational physicians should be sensitized and pay more attention to these aspects.

## 8. Study Limitations

The low number of subjects, self-selection of the respondents, and the risk of conformist attitude limit the analysis. Moreover, questionnaires filled out by email generate both benefits (comfort for the respondent, reduced influence of the interview operator) and disadvantages (lower response rate, lack of representativeness caused by respondents’ self-selection). Additionally, no personality traits were analyzed, with a relevant variable missing. This study is a cross-sectional survey, and despite assessing associations between outcomes and different variables, it does not allow to support the causal relationship between them. Predictive factors found in this study and the effects of changes over time need to be assessed in future research. Considering that personality plays an important causal role in burnout development, individual characteristics should be considered in the future. Moreover, external validation of a larger sample of physicians would add value to the results.

## 9. Conclusions

An important percentage of SIPhs are suffering from work-related burnout. Burnout was reported not only as a stress-induced syndrome but as a low self-esteem-induced syndrome, too.

Identifying predictive factors allow for a personalized and evidence-based approach to disease prevention. Raising awareness of the disease triggers makes early intervention possible and improves management effectiveness.

It is expected that promoting social insurance physicians’ role in the social security system and value their knowledge, responsibilities, and actions to strengthen their position in relation to public officials and other medical specialties.

Modifying changeable mainly professional risk factors and promoting professional help would bring benefits to health professionals. Good quality of work life increases engagement, efficiency, and results.

## Figures and Tables

**Table 1 medicina-59-00436-t001:** Sociodemographic and professional characteristics of participants.

Characteristics	Number/% (±SD)
Mean age (years)	48.86 ± 7.94
Gender (female)	56 (75.7%)
Marital status (yes)	60 (81.08%)
Children (yes)	58 (78.38%)
Vacations (average days)	18.43 ± 8.47
Sick leave (average days)	5.95 ± 16.85
Psycho-traumatic events (yes)	26 (35.14%)
Mean length of service duration (years)	17.09 ± 8.49
Daily commute (yes)	24 (32.43%)
Second job (yes)	26 (35.14%)
Workload (high or excessive)	54 (72.97%)

**Table 2 medicina-59-00436-t002:** Correlation of different variables with burnout syndrome.

Variable	Correlation Coefficient	Sig. (Two-Tailed)—*p*
Number of days off	−0.31	=0.007
Being in a couple	−0.33	=0.004
Psychotrauma	0.24	=0.037

**Table 3 medicina-59-00436-t003:** The backward selection type of multiple logistic regression with predictive factors for burnout (R Square = 0.22).

Variable	B Coefficient	Standard Error	Significance (*p*)
Constant	1.89	0.17	<0.001
Days off	−0.12	0.05	=0.023
Being in a couple	−0.29	0.14	=0.036
Trauma	0.22	0.11	=0.041

**Table 4 medicina-59-00436-t004:** Burnout side-effects.

Burnout Indices	Absence of Burnout (Number)	*p* Value
Hostility	28 (60.87%)	=0.010
Professional routine	28 (60.87%)	<0.001
Lack of career development	36 (78.26%)	<0.001
Inefficiency	28 (60.87%)	<0.001
Reduced organization	40 (86.96%)	<0.001
Altered family relationships	26 (56.52%)	=0.016
Cognitive distortions with depression	46 (100%)	=0.002
Poor performance at work	38 (82.61%)	<0.001

**Table 5 medicina-59-00436-t005:** Percentage of depersonalization.

Depersonalization	Low	Average	High
Persons	48 (65%)	10 (13%)	16 (22%)
Mean (±SD)	2.5 (±2.08)	9	15.75 (±1.53)

**Table 6 medicina-59-00436-t006:** Correlation of different factors with depersonalization.

Variable	Correlation Coefficient	Sig. (Two-Tailed)—*p*
Daily commute	−0.27	=0.021
Gender	0.24	=0.037
Children	0.39	=0.001

**Table 7 medicina-59-00436-t007:** Backward selection type of multiple logistic regression for depersonalization.

Variable	B Coefficient	Standard Error	Significance (*p*)
Constant	1.09	0.12	<0.001
Children	0.43	0.12	=0.001
Daily commute	−0.25	0.11	=0.022

**Table 8 medicina-59-00436-t008:** Inducing factors for burnout.

Inducing Factors for Burnout	*p* Value
Professional factors	
Taking on additional tasks Deadline pressure High workload Low efficiency	=0.001 <0.001 =0.003 =0.009
Personal factors	
Extended family Personal surroundings	=0.002 =0.007

**Table 9 medicina-59-00436-t009:** Backward selection type of multiple logistic regression for burnout contributing factors.

Variable	B Coefficient	Standard Error	Significance (*p*)
Constant	0.95	0.08	<0.001
Deadline pressure	0.23	0.04	<0.001
High workload	0.10	0.04	=0.012

**Table 10 medicina-59-00436-t010:** Coping mechanisms.

Coping Mechanisms	Subscale	Low Burnout	Average/High Burnout	*p*-Value
**Problem-based methods**	
- Making a plan of action - Doing one step at a time	Planning Active coping	42 (91.30%) 42 (91.30%)	20 (71.43%) 18 (64.29%)	=0.029 =0.006
**Emotion-based methods**	
- Seeking emotional support from fellows - Accepting what has happened	Emotional and social support Acceptance	34 (73.91%) 32 (69.57%)	12 (42.86%) 10 (35.71%)	=0.022 =0.014
**Non-functional**	
- Turning to other substitute activities - Going to movies / watching TV - Expressing emotional distress	Mental disengagement Mental disengagement Emotional discharge	40 (86.96%) 34 (73.91%) 24 (52.17%)	14 (50%) 10 (35.71%) 12 (42.86%)	=0.004 =0.001 =0.012

**Table 11 medicina-59-00436-t011:** The backward selection type of multivariate logistic regression model with coping strategies for burnout (Nagelkerke R Square = 0.34).

Coping Strategies	B Coefficient	Standard Error	Significance (*p*)
Doing one step at a time	−0.16	0.08	=0.041
Accepting what has happened	−0.17	0.05	=0.001
Going to movies/watching TV	−0.11	0.05	=0.023
Constant	2.44	0.21	<0.001

**Table 12 medicina-59-00436-t012:** Gender-dependent coping mechanisms.

Coping Mechanisms	Subscale	Women	Men	*p*-Value
**Problem-based methods**	
- Making a plan of action - Doing one step at a time	Planning Active coping	50 (89.29%) 48 (85.71%)	12 (66.67%) 12 (66.67%)	=0.034 =0.012
**Emotion-based methods**	
- Seeking emotional support from fellows - Accepting what has happened	Emotional and social support Acceptance	40 (71.43%) 36 (64.29%)	6 (33.33%) 6 (33.33%)	=0.009 =0.003
**Non-functional**	
- Turning to other substitute activities - Going to movies/watching TV - Expressing emotional distress	Mental disengagement Mental disengagement Emotional discharge	44 (78.56%) 38 (67.86%) 32 (57.14%)	10 (55.56%) 6 (33.33%) 4 (22.22%)	=0.032 =0.005 =0.035

## Data Availability

The data presented in this study are available on reasonable request from the corresponding author. The data are not publicly available due to privacy and ethical reasons.

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
