# Peer review of "Social Insurance Physician Burnout—Stress Factors and Coping Strategies"

_medicina, 2023, doi:10.3390/medicina59030436_

Round 1
Reviewer 1 Report
The study includes the minimum standards for article acceptance. The abstract and the introduction include a very good theoretical foundation. the results and discussions are very good and the references reflect the current research. Congratulations to the authors.
Author Response
Dear member of the reviewer board,
Thank you for taking the time to read our article proposed for publication and thank you for your kind words.
Sincerely,
Dr. Corina Oancea.
Reviewer 2 Report
Thank you for the opportunity to review Oancea and colleagues' paper entitled "Social insurance physician burnout- stress factors and coping strategies." I have major and minor comments to help improve the quality of the paper.
P2: The entire introduction needs more depth, and research gaps were not highlighted. I would suggest that you modify it and improve the logical flow. More justification must be added on why social insurance physicians were chosen for the study. It was only briefly discussed in the background and was not strong enough to convince the readers. What were the stressors they experienced published in previous studies?
P2: Add more explanation on why MBI was chosen to measure burnout.
P3: Magnitude of the exhaustion syndrome? Do you mean the impact of the exhaustion syndrome?
P3: How was the research purpose explanation for the participants conducted since the data collection method was implemented through questionnaires via email? Was it in-person or online via Zoom?
P3: How was the shortened version of scale MBI-HSS translated in Romanian validated? Was it already previously implemented in other published research?
P3: You mentioned the MBI-HSS scale was translated into Romanian; however, the country where the research was conducted was unclear. Was this research conducted in Romania?
P4:You surveyed 74 social insurance physicians. However, the sample size is not justified. Where did you base these 74 respondents? What is the justification for this value?
P4: What is depersonalization? It may need to be clarified to the readers.
P13: You mentioned geographic region of origin and cultural specificity in coping strategies; however, its role should have been discussed regarding burnout.
P13: The first paragraph should focus on the study's main findings to improve the discussion. The authors did it; however, the interpretation of the findings needs to be improved for more clarity for the readers. Next, you should talk about your specific findings, and the logical flow should be from specific to more general ones. Your topic sentence should be your particular finding and not the finding of other studies.
Author Response
Dear member of the reviewer board,
Thank you for taking the time to read our article proposed for publication.
Thank you for your pertinent remarks. I tried to answer them and introduced some changes in the document, as mentioned below.
Sincerely,
Dr. Corina Oancea.
P2: The entire introduction needs more depth, and research gaps were not highlighted. I would suggest that you modify it and improve the logical flow.
P1-2. Differences in study designs, burnout definition and assessments methods were also found. This information highlights the impossibility of obtaining clear figures on the prevalence of burnout and emphasize the importance of reaching an agreement on the definition of burnout and the standardization of measurement tools to assess the effects of chronic occupational stress on physicians. (3)
More justification must be added on why social insurance physicians were chosen for the study. It was only briefly discussed in the background and was not strong enough to convince the readers. What were the stressors they experienced published in previous studies?
P2. In European social security organizations, several countries have reported shortages of social insurance physicians (45) and Romania is no exception. According to the national records, around 30% of SIPhs have left the national network or reduced their activity in the last 15 years.
Several hypotheses have been discussed for this lack. With a significant number of physicians leaving the system, the amount of work has been transferred and increased for those who remain, until a viable long-term solution is found, including attracting other specialists into the system. Like their European colleagues, Romanian SIPhs frequently reported organizational problems and computer system deficiencies. Other aspects were also considered besides the increased demand for medical evaluations: a greater focus on professional reintegration, case assessments became increasingly demanding and time-consuming due to the shift to more complex or migrant cases (46).
The medical assessments for short-term (sickness) and long-term (disability) benefits have traditionally been undertaken by social insurance physicians. (47) Other tasks were also described and these vary considerably between countries. In Romania SIPhs also coordinate the return to work program and some participate in appeals commissions. (14)
- de Wind, A. E., Brage, S., Latil, F., & Williams, N. (2020). Transfer of tasks in work disability assessments in European social security. European Journal of Social Security, 22(1), 24–38. https://doi.org/10.1177/1388262720910307
- 46. https://www.eumass.eu/wp-content/uploads/2019/03/Report-1-2019-1.pdf , accessed 15 Feb 2023
- de Boer W.E.L., Brenninkmeijer V., Zuidam W. (2004) ‘Long-term disability arrangements. A comparative study of assessment and quality control’, TNO report, TNO, Hoofddorp; https://repository.tno.nl//islandora/object/uuid:b53d3ea3-247e-4ea0-9ad6-df81b1f0cf98
P2: Add more explanation on why MBI was chosen to measure burnout.
A: (P2). The most widely used measure of burnout is the Maslach Burnout Inventory, which has high reliability and validity. (30) As shown by Rotenstein et al in a systematic review, 85.7% of studies used a version of the Maslach Burnout Inventory to assess burnout. (3)
P3: Magnitude of the exhaustion syndrome? Do you mean the impact of the exhaustion syndrome?
P3. The extent of the exhaustion syndrome
P3: How was the research purpose explanation for the participants conducted since the data collection method was implemented through questionnaires via email? Was it in-person or online via Zoom?
P3. It was explained in writing, through an information letter attached to the email together with the survey
P3: How was the shortened version of scale MBI-HSS translated in Romanian validated? Was it already previously implemented in other published research?
P4. The study was carried out with the aim of characterizing this population group and testing the possible response rate, by reducing the number of questions and offering anonymity.
Following a comprehensive evaluation and validation by statistical analysis, we proposed a questionnaire-based survey to evaluate stressors and coping strategies for social insurance physicians, with the following items: socio-demographic data, short MBI-HSS, brief COPE and a set of questions focused on the potential inducing factors for burnout.
Thus, a complex assessment tool was obtained, easy to apply and with reduced costs. However, the results need an external validation on a larger sample of physicians, adding a second related specialty, occupational medicine and by referring to other validated questionnaires.
P2-3. The MBI-HSS is translated, validated (48) and widely used (49) in Romania in different studies and some researchers made individual analysis of its items (50) or used adapted MBI-HSS (51,52) The value of the MBI-HSS shorten version scale, its ability to capture the burnout phenomenon was not a goal pursued in this article. The score's properties (validity, reliability) can be discussed in a future article.
- Raulea C. Prevenirea sindromului burnout in organizatiile romanesti – un studiu pilot în rândul personalului medical, Antropomedia 2010; I(2):91-100, ISSN 2067-6107
- Popa F, Raed A, Purcărea VL, Lală A, Bobirnac G. Occupational Burnout levels in Emergency Medicine – a nationwide study and analysis. Journal of Medicine and Life 2010;3(3)
- Bria M - Sindromul burnout în rândul personalului medical, PhD thesis, Babeş-Bolyai University, Cluj-Napoca, Romania, 2013
- Sirghie RE – Surse de distres profesional si particularitati de teren psihic la personalul medical ATI (vulnerabilitatea la stres vs. Rezilienta), PhD thesis, Carol Davila University, Bucharest, Romania, 2017
- Dumea, E. ., Efrim, N. D., Petcu, A. ., Anghel, L., & Puscasu, C. G. . (2022). Burnout Syndrome in Personnel of an Infectious Diseases Hospital, One Year after the Outbreak of the COVID-19 Pandemic. BRAIN. Broad Research in Artificial Intelligence and Neuroscience, 13(1Sup1), 230-246. https://doi.org/10.18662/brain/13.1Sup1/316
In addition at the study limitations:
Also, an external validation on a larger sample of physicians would add value to the results
P3: You mentioned the MBI-HSS scale was translated into Romanian; however, the country where the research was conducted was unclear. Was this research conducted in Romania?
P1. The research was conducted in Romania.
P4: You surveyed 74 social insurance physicians. However, the sample size is not justified. Where did you base these 74 respondents? What is the justification for this value?
P4-5. Seventy four social insurance physicians voluntarily participated at the study.
P4: What is depersonalization? It may need to be clarified to the readers.
P6. Depersonalization (professional detachment) refers to the adoption of negative and indifferent attitude toward others.
P13: You mentioned geographic region of origin and cultural specificity in coping strategies; however, its role should have been discussed regarding burnout.
P10. There were described relationships between culturally learned values, beliefs, attitudes and consecutive behavioral patterns. This concept refers to the fact that individuals operate and behave accordingly to traditionally dictated attitudes and beliefs about how the life should be lived, including the manner in which one copes with life’s problems. (44) Cultural norms and family characteristics influence their ability to function. Traditional large families of certain ethnic communities facilitate seeking social support from extended family members (44) while in in societies strongly influenced by religion, this becomes the most frequently used stress-coping strategy. (43)
P13: The first paragraph should focus on the study's main findings to improve the discussion. The authors did it; however, the interpretation of the findings needs to be improved for more clarity for the readers. Next, you should talk about your specific findings, and the logical flow should be from specific to more general ones. Your topic sentence should be your particular finding and not the finding of other studies.
P8-11. We improved the discussion section.
Reviewer 3 Report
The paper shows a sound study of the incidence of burnout in a sample of social insurance Physicians or medical advisers, who suffer pressure from both insurer and clients, also indicating ways of preventing burnout at both an individual and institutional level. However, some points must be attended in my opinions:
1) The authors say that appropiate coping strategies may increase individual resistance [19]. The relationship between these strategies and developing resilence – as a more personalized way of preventing stress – could also be mentioned. It would also be appropiate to cite some paper dealing with the relationship between resistance and despersonalization.
2) If it is possible, it would be interesting to cite some result (either from the data obtained in the present study or from other studies) which links despersonalization to the workplace clima, specially regarding the quality of interpersonal relationships.
The aim of this suggestions is to know if it is possible to contribute to an improvement of individual wellbeing of the physicians through the improvement of the interpersonal relationships in the working personal environment.
Author Response
Dear member of the reviewer board,
Thank you for taking the time to read our article proposed for publication.
Thank you for your pertinent remarks. I tried to answer them and introduced some changes in the document, as mentioned below.
Sincerely,
Dr. Corina Oancea.
1) The authors say that appropiate coping strategies may increase individual resistance [19]. The relationship between these strategies and developing resilence – as a more personalized way of preventing stress – could also be mentioned. It would also be appropiate to cite some paper dealing with the relationship between resistance and despersonalization.
P3. Conversely, an inappropriate way to deal with high-level stress is for employees to psychologically withdraw from others, leading to dehumanized and depersonalized contact. This dysfunctional coping strategy will lead to further deterioration of the relationship with others. (53,54)
- De Clercq, D., Haq, I., & Azeem, M. The relationship between workplace incivility and depersonalization towards co-workers: Roles of job-related anxiety, gender, and education. Journal of Management & Organization 2020;26(2):219-240. doi:10.1017/jmo.2019.76
- Filip Van Droogenbroeck, Bram Spruyt, Christophe Vanroelen. Burnout among senior teachers: Investigating the role of workload and interpersonal relationships at work. Teaching and Teacher Education 2014; 43:99-109, ISSN 0742-051X. doi.org/10.1016/j.tate.2014.07.005.
2) If it is possible, it would be interesting to cite some result (either from the data obtained in the present study or from other studies) which links depersonalization to the workplace climate, especially regarding the quality of interpersonal relationships.
P2. Job satisfaction was found as a negative predictor of each type of burnout subscale by some authors, while conflict was a factor contributing positively both to emotional exhaustion and depersonalization scores. (55)
- Piko BF. Burnout, role conflict, job satisfaction and psychosocial health among Hungarian health care staff: a questionnaire survey. Int J Nurs Stud. 2006;43(3):311-8. doi: 10.1016/j.ijnurstu.2005.05.003. Epub 2005 Jun 16. PMID: 15964005.
The aim of this suggestions is to know if it is possible to contribute to an improvement of individual wellbeing of the physicians through the improvement of the interpersonal relationships in the working personal environment.